# Cytokines and Pathogenesis of Central Retinal Vein Occlusion

**DOI:** 10.3390/jcm9113457

**Published:** 2020-10-27

**Authors:** Hidetaka Noma, Kanako Yasuda, Masahiko Shimura

**Affiliations:** Department of Ophthalmology, Hachioji Medical Center, Tokyo Medical University, 1163, Tatemachi, Hachioji, Tokyo 193-0998, Japan; kana6723@yahoo.co.jp (K.Y.); masahiko@v101.vaio.ne.jp (M.S.)

**Keywords:** cytokines, central retinal vein occlusion, macular edema, neovascularization

## Abstract

Central retinal vein occlusion (CRVO) causes macular edema and subsequent vision loss and is common in people with diseases such as arteriosclerosis and hypertension. Various treatments for CRVO-associated macular edema have been trialed, including laser photocoagulation, with unsatisfactory results. However, when the important pathogenic role of vascular endothelial growth factor (VEGF) in macular edema was identified, the treatment of CRVO was revolutionized by anti-VEGF therapy. However, despite the success of intraocular injection of anti-VEGF agents in many patients with CRVO, some patients continue to suffer from refractory or recurring edema. In addition, the expression of inflammatory cytokines increases over time, causing more severe inflammation and a condition that is increasingly resistant to anti-VEGF therapy. This indicates that the pathogenesis of macular edema in CRVO is more complex than originally thought and may involve factors or cytokines associated with inflammation and ischemia other than VEGF. CRVO is also associated with leukocyte abnormalities and a gradual reduction in retinal blood flow velocity, which increase the likelihood of it developing from the nonischemic type into the more severe ischemic type; in turn, this results in excessive VEGF expression and subsequent neovascular glaucoma. Here, we review the role of different factors and cytokines involved in CRVO pathogenesis and propose a mechanism that holds promise for the development of novel therapies.

## 1. Introduction

Central retinal vein occlusion (CRVO) can result in vision loss, the main cause of which is macular edema. Several therapeutic approaches, including retinal photocoagulation, were trialed with little success [1] but once vascular endothelial growth factor (VEGF) was recognized as playing a central role in CRVO-associated macular edema, treatment of the condition was revolutionized by anti-VEGF agents that could be injected into the eye [2]. However, macular edema is recurrent or resistant to anti-VEGF therapy in some patients, suggesting that other factors are involved in its pathogenesis. In addition, diseases such as type 2 diabetes, hyperlipidemia, and hypertension, which are increasing in prevalence due to a rise in unhealthy eating habits and an aging population, are among the significant risk factors for CRVO. This development highlights the need to better understand the pathogenesis of CRVO and to identify novel therapeutic approaches. To this end, we review the pathogenesis of CRVO, exploring the roles of VEGF and other factors.

## 2. Pathogenesis of CRVO

We previously reviewed the pathogenesis of branch retinal vein occlusion (BRVO) [3], but the pathogenesis of CRVO is different from that of BRVO [4].

Like BRVO, CRVO can be nonischemic or ischemic, depending on the degree of retinal hypoxia (Figure 1a,b) [4]. CRVO occurs mainly in older people when the central retinal vein is compressed by an arteriosclerotic central retinal artery in the vicinity of the lamina cribrosa, resulting in venous thrombosis [5]. Bleeding in the nerve fiber layer in CRVO causes radial brush-shaped splinter hemorrhages. Cystoid spaces are formed as blood components leak into the retina and accumulate in the outer plexiform layer and internal granular layer. However, CRVO may also occur in younger people without any comorbidity; the pathogenesis of these cases is unclear, but case reports and series have proposed inflammation, exercise and dehydration, and congenital anomalies as potential explanations [6,7,8].

The macula of the retina gives us detailed vision. The small pit in its center, the fovea, which consists entirely of cones, is responsible for the sharpest visual acuity [9]. CRVO-related macular edema is thought to be caused by the blood–retinal barrier (BRB) breaking down. Other aspects are involved in macula edema, such as hydrostatic and osmotic pressure, which also lead to fluid accumulation and tissue swelling [10]. The cause of this is damage to the tight junctions between capillary endothelial cells [11], coupled with vitreomacular traction and adhesion [12]. This breakdown of the BRB results in vasopermeability factors, including VEGF, leaking into the vitreous fluid [13].

Progression of ischemia in CRVO can result in neovascular glaucoma, an aggressive condition that causes rapid vision loss and has a poor prognosis. Both BRVO and CRVO can cause vitreous hemorrhage and rubeotic glaucoma: these sequelae are more common in CRVO and can lead to blindness [4], but they may also occur in BRVO, depending on the area and severity of ischemia. Nevertheless, generally, CRVO has a worse outcome than BRVO [14]. Therefore, CRVO is a potentially serious retinal disease that can lead to severe visual impairment.

## 3. VEGF and Macular Edema

Retinal pigment epithelial cells, retinal glial cells, and vascular endothelial cells all express VEGF [15]. Upregulation of VEGF expression is induced by hypoxia and results in increased endothelial cell proliferation, vascular permeability, and angiogenesis [16]; it causes angiogenesis by facilitating the rearrangement of actin filaments in the cytoplasm. VEGF also increases the phosphorylation of tight junction proteins, including zonula occludens-1 and occludin [17].

Evidence that VEGF is involved in CRVO-related macular edema came from our previous study of vitreous fluid samples taken during vitreous surgery to treat macular edema [18]. We detected VEGF in the vitreous fluid of patients with CRVO at a significantly higher level than in control patients with nonischemic conditions such as a macular hole [18]. These findings led to VEGF inhibition being trialed as a treatment for macular edema in people with CRVO, and a marked improvement was observed. On the other hand, hypoxia-induced VEGF was reported to be the linking factor between retinal ischemia and retinal neovascularization in CRVO [19]. However, VEGF levels vary widely between patients, and they are significantly lower in patients with CRVO without retinal ischemia than in those with retinal ischemia [18]. We, therefore, focused our attention on the nonperfusion area as the potential cause. Dividing our CRVO patient cohort into a nonischemic group and an ischemic group, based on the degree of retinal hypoxia, revealed that the intraocular concentration of VEGF was significantly higher in patients with ischemic CRVO than in those with nonischemic CRVO (Figure 2) [18]. The severity of macular edema was also directly related to the level of VEGF [18]. These findings suggested that retinal ischemia in CRVO causes an initial increase in VEGF due to vascular occlusion, which disrupts the BRB, causing macular edema.

VEGF also increases the vascular permeability of the eye. The level of VEGF increases with the severity of retinal ischemia [13], leading to more neovascularization and potentially resulting in vitreous hemorrhage and neovascular glaucoma. Therefore, the pathogenesis of CRVO seems to be strongly associated with retinal ischemia.

## 4. Inflammation and Other Putative Mechanisms of Macular Edema in CRVO

Despite the above evidence, macular edema can occur even in cases of CRVO with mild retinal ischemia (Figure 1c,e) or in nonischemic CRVO. Furthermore, anti-VEGF therapy is more effective for nonischemic CRVO than ischemic CRVO [20], and even CRVO with mild retinal ischemia can result in relatively high VEGF levels [18]. As ischemic CRVO causes severe cytopathy, the prognosis for regaining visual acuity is poor, even if anti-VEGF therapy is administered. Additionally, VEGF levels are not particularly high in nonischemic CRVO, and VEGF can be suppressed by anti-VEGF therapy. These observations indicate that the pathogenesis of CRVO-related macular edema cannot be fully explained by retinal ischemia. Similarly, some patients with CRVO and macular edema do not respond to anti-VEGF therapy [21], indicating that its etiology may involve other factors and cytokines. Intravitreal injections of steroids are an effective treatment, suggesting that inflammation plays a role [22]. Interestingly, inflammatory cytokines, intraocular levels of growth factors, chemokines, adhesion molecules, and other factors are increased in CRVO patients with macular edema [23,24,25,26,27]. These are described in more detail below.

### 4.1. Cytokines

#### 4.1.1. Interleukin 6

The inflammatory cytokine interleukin (IL)-6 has several functions, including effecting actin filament rearrangement, which can create gap junctions between cells and increase endothelial permeability [28]. Patients with CRVO with ischemia had significantly higher vitreous fluid levels of IL-6 than patients with CRVO without ischemia, and IL-6 levels correlated significantly with the extent of macular edema [18]. IL-6 plays a role in protecting retinal ganglion cells [29,30]; furthermore, it is hypothesized to mediate the inflammation and remodeling of retinal blood vessels induced by angiotensin II and be involved in axonal degeneration in glaucoma [30,31]. IL-6 mRNA expression increases over time in hypoxia-exposed cultured endothelial cells [32,33,34].

#### 4.1.2. Interleukin 8

Exposing vascular endothelial cells to hypoxia and oxidative stress also induces the production of IL-8, a potent chemoattractant that also activates neutrophils and T-cells [35,36,37]. IL-8 is expressed in the retina, is activated in various ocular diseases, including CRVO and glaucoma, and may play an important role in the activation of other cytokines [26,38,39]. IL-8 regulates endothelial permeability by downregulating tight junction proteins [40,41]. It also increases leukocyte adhesion to the vascular endothelium [42,43].

### 4.2. Growth Factors

#### 4.2.1. Placental Growth Factor

Placental growth factor (PlGF), a 25-kd dimeric protein, is a member of the VEGF family and is highly homologous with VEGF [44,45]. In nonischemic CRVO, PlGF may act together with sICAM-1 and IL-8 to reduce the relative blood flow velocity, measured as the mean blur rate (MBR) [46]. A specific ligand for VEGFR-1, PlGF potently promotes angiogenesis and stimulates endothelial cell growth and migration [47]. It also binds exclusively to VEGFR-1 [48], with a higher affinity than VEGF [49]. PlGF stimulates tissue factor production and chemotaxis in monocytes and macrophages, thus modulating the inflammatory process [50].

#### 4.2.2. Platelet-Derived Growth Factor

Platelet-derived growth factor (PDGF) is a dimeric glycoprotein that belongs to the PDGF/VEGF family and helps regulate the migration of mesenchymal cells (fibroblasts, smooth muscle cells, and glial cells). In CRVO, the concentration of PDGF-AA is significantly higher in the aqueous humor, and PDGF-AA is hypothesized to contribute to inflammation [51]. PDGF is involved in the formation of blood vessels and astrocyte recruitment during the development of the retina [52,53]. When arteriosclerosis decreases blood flow, expression of PDGF-AA (PDGF composed of two A subunits) increases in endothelial cells [54]. PDGF-AA acts on the induction of the gap bond formation [55].

### 4.3. Monocyte Chemoattractant Protein

The chemokine monocyte chemoattractant protein (MCP-1) promotes monocyte chemotaxis and the phosphorylation of tight junction proteins (e.g., zonula occludens-1 and occludin) [56,57]. MCP-1 correlates with morphologic changes, such as serious retinal detachment in CRVO [58]. Retinal ischemia, arteriosclerosis, and oxidative stress all increase the expression of MCP-1 [59,60,61].

### 4.4. Intercellular Adhesion Molecule 1

Intercellular adhesion molecule 1 (ICAM-1) is an adhesion molecule expressed in various cells in the retina and choroid, including normal retinal pigment epithelium cells and leukocytes [62]. Retinal ischemia upregulates the expression of ICAM-1 mRNA and protein [63,64], which causes increased leukocyte rolling and adhesion to vessel walls, leading to leukostasis and blood stagnation [65]. Similar effects are observed in retinal vein occlusion [66]. Thus, CRVO-induced leakage from the retinal microvessels could be further increased by leukocyte entrapment as they adhere to vascular endothelial cells.

### 4.5. Interferon-Inducible 10-kDa Protein

Interferon-inducible 10-kDa protein (IP-10) exerts an antiangiogenic effect by inhibiting the proliferation of endothelial cells and inducing their apoptosis [67]. Levels of IP-10 are elevated in the vitreous fluid in retinal vein occlusion, as well as in proliferative diabetic retinopathy and rhegmatogenous retinal detachment [39]. IP-10 also inhibits VEGF-induced endothelial motility [68]. It is a CXC chemokine that is secreted by macrophages, endothelial cells, and fibroblasts. It acts as a chemoattractant for macrophages, dendritic cells, and T-cells, contributes to T-helper type 1 immune responses, and activates cell-mediated immunity in general.

### 4.6. Pentraxin 3

Pentraxin (PTX) 3 is an acute-phase protein that is involved in innate immunity and inflammation. PTXs are a family of acute-response proteins consisting of three members: C-reactive protein (CRP), serum amyloid P, and PTX3. PTX3 is produced by retinal pigment epithelial cells in response to proinflammatory cytokines and may mediate the retinal inflammatory response [69]. Park et al. [70] found that levels of PTX3 are significantly higher in patients with CRVO and BRVO than in controls and proposed that PTX3 may be a suitable biomarker for diagnosing retinal vein occlusion. One study reported that hypertensive patients with elevated serum CRP levels may be more at risk of developing CRVO [71]. These are classic acute-phase reactants that reflect inflammatory activity [72,73]. While CRP and serum amyloid P are mainly synthesized in the liver [74,75], PTX3 is mainly produced by cells in extrahepatic tissues, including vascular endothelial cells and fibroblasts, and is induced by cytokines [72,76,77,78,79,80]. Overexpression of PTX3 results in impaired survival after intestinal ischemia and reperfusion, manifesting as tissue damage and death [81]. PTX3 also modulates leukocyte recruitment in inflammation [82], so vascular permeability could increase if PTX3 is upregulated [83].

### 4.7. Erythropoietin

Erythropoietin (EPO) is known to be regulated by the oxygen-sensitive transcription factor Hifl*α* [84]. In addition to its well-established hematopoietic effects, EPO has neuroprotective properties in the retina [85]. Studies reported that ocular EPO levels are higher than normal in patients with CRVO [86,87]. In animal models, EPO was found to be associated with retinal ischemia and to have neuroprotective effects against ischemia-reperfusion injury and light-induced retinal degeneration [88]. Higher vitreal levels of EPO were found to be correlated with higher vitreal levels of VEGF and more severe macular edema [86].

### 4.8. Other Putative Mechanisms

In addition to the changes in cytokines and inflammatory mediators discussed above, patients with CRVO have higher values of aqueous flare, an index of ocular inflammation, than control patients [89], providing further evidence of active inflammation in CRVO. Furthermore, the grade of aqueous flare is positively correlated with the intraocular levels of inflammatory cytokines [89]; interestingly, this correlation is stronger in cases of CRVO than in BRVO [89,90], suggesting that inflammation is more severe in patients with CRVO, which may be due to the extensive retinal hemorrhaging that occurs in CRVO. Basic [25,27] and clinical [28,40,41,56,57] studies also show that levels of the abovementioned factors/cytokines, as well as VEGF, are significantly higher in ischemic CRVO than in nonischemic CRVO.

These findings indicate that CRVO causes ischemia and inflammation. Specifically, we propose that the pathogenesis of ischemic CRVO involves both ischemia and inflammation, whereas nonischemic CRVO mainly involves inflammation. This leads to an increase in the expression of VEGF and inflammatory cytokines, BRB disruption, and the onset and progression of macular edema. We have outlined this proposed mechanism in Figure 3.

## 5. VEGF Receptors and Macular Edema

Since VEGF increases vascular permeability, it is likely a key factor in macular edema development. VEGF exerts its effects via signaling pathways activated upon its binding to VEGF receptors. In the retina, where VEGFR-1 and VEGFR-2 are expressed, VEGF binding to either receptor activates autophosphorylation [91,92,93] and transphosphorylation, and a signaling cascade is initiated. Patients with CRVO have significantly higher levels of VEGFR-1 and VEGFR-2 in the aqueous humor than patients with cataracts but without CRVO [51], which points to the involvement of these receptors in the pathogenesis of CRVO-associated macular edema.

Activation of VEGFR-1, which is mainly expressed by monocytes/macrophages, promotes inflammation in a number of ways [50,94,95]. Signaling via this receptor contributes to the recruitment of leukocytes to sites of inflammation [96]. The VEGF family member PlGF binds specifically to VEGFR-1, stimulating tissue factor production and monocyte/macrophage chemotaxis [50]. Tissue factor is involved in activating the extrinsic coagulation cascade pathway [97]. In cultured monocytes, the binding of PlGF to VEGFR-1 increases proinflammatory factor production via a calcineurin-dependent pathway [98], indicating that this molecule directly influences the inflammatory response.

In contrast, binding of VEGF to VEGFR-2, which is expressed exclusively by endothelial cells, initiates signaling that increases vascular permeability and upregulates inflammatory cytokine expression (such as MCP-1 and ICAM-1) via NF-κB [99,100,101]. These cytokines induce leukocyte chemotaxis and inflammatory cell adhesion to the vascular endothelium, which further increases vascular permeability. Thus, VEGF binding to its receptors on vascular endothelial cells, monocytes, and macrophages causes macular edema to develop and progress. Vascular permeability in the inflammatory response might also be influenced by the soluble, circulating forms of these VEGF receptors, sVEGFR-1 and sVEGFR-2. Via these receptors, VEGF acts as a chemotactic factor for inflammatory cells [102]. Together, these lines of evidence indicate that VEGF promotes inflammation as well as increasing vascular permeability.

## 6. Inflammation and Retinal Blood Flow Velocity

Retinal blood flow is slower in patients with CRVO than in people with healthy eyes [103,104,105]. In our previous study, we identified moving particles, considered to be leukocytes, in capillary vessels when we measured retinal blood flow velocity using scanning laser ophthalmoscopy with fluorescein angiography [106,107]. During inflammation, leukocytes accumulate in the lesion and the velocity of local blood flow decreases. We compared retinal blood flow velocity in healthy individuals and patients with BRVO or CRVO and found that the more severe the retinal vein occlusion, the lower the blood flow velocity [108]. Blood flow velocity was also negatively correlated with the level of sICAM-1 in the vitreous fluid in patients with CRVO, indicating that the damage to the vascular endothelium had increased the expression of adhesion factors and resulted in enhanced leukocyte rolling and adhesion and a consequent reduction in blood flow velocity [109]. Furthermore, blood flow in the large vessels in and around the optic disk in patients with CRVO, measured using laser speckle flowgraphy to determine the MBR, was negatively correlated with the log-transformed levels of PlGF, sICAM-1, and IL-8 in the aqueous humor [46]. These findings appear to indicate blood flow velocity in patients with CRVO may be reduced as a result of the increased rolling, chemotaxis, adhesion, and entrapment of leukocytes caused by elevations in these inflammatory and growth factors (Figure 3). Blood flow velocity also worsens as retinal edema increases, resulting in longer diffusion distances and creating a positive feedback loop that further worsens ischemia.

In contrast to the above findings, other studies have provided evidence that rather than inflammation causing reduced blood flow, reduced blood flow due to stagnation and very low shear stress may induce a proinflammatory state, resulting in CRVO [110,111]. The reduced flow may induce inflammation via mediators such as Krüppel-like factors and nuclear factor kappa-light-chain-enhancer of activated B-cells (NFKb) [112]. Several studies have shown that static blood flow induced inflammation, leukocyte adhesion, and barrier disruption, even in retinal endothelial cells [113,114].

An alternative hypothesis is that inflammation itself causes macular edema and that CRVO is a consequence of this edema.

To summarize, the role of inflammation and retinal flow in macular edema and CRVO remains unclear and requires further study.

## 7. Proposed Mechanism of CRVO Pathogenesis

In summary, we propose that CRVO causes inflammation as a result of retinal ischemia and hemorrhage, which increases the expression of VEGF and other inflammatory factors in the eye and disrupts the BRB, thus causing the onset and progression of macular edema. It is also likely that VEGF and PlGF recruit leukocytes and upregulate inflammatory cytokines by their action on VEGFR-1. In addition, VEGFR-2 activation by VEGF increases vascular permeability and further expression of inflammatory cytokines, including MCP-1 and ICAM-1 via NF-κB. This results in leukocyte chemotaxis and adhesion to the vascular endothelium, which further reduces local blood flow velocity, creating a positive feedback loop in which ischemia continues to increase [115]. A second positive feedback loop is established by the increased inflammation that arises as a result of leukocyte chemotaxis and adhesion. We hypothesize that late-stage, chronic CRVO is associated with greater inflammation than early-stage CRVO and has a more complicated pathological mechanism. This mechanism includes macular edema becoming refractory as it develops into chronic macular edema as the hemorrhaged blood is gradually absorbed; this process is the result of the positive feedback loop that causes levels of VEGF and inflammatory cytokines, such as MCP-1 and ICAM-1, to steadily increase. This, in turn, leads to leukocyte abnormalities and a slower retinal blood flow, increasing the risk of CRVO becoming ischemic (Figure 4). Indeed, the Central Vein Occlusion Study showed that CRVO was ischemic in approximately 20% of patients [116,117,118]. Corticosteroids may suppress positive feedback and prevent the transition from nonischemic to ischemic CRVO.

VEGF levels become even more excessive as hypoxia increases [13], ultimately resulting in vitreous hemorrhage and neovascular glaucoma. The RAVE trial demonstrated that intravitreal administration of the anti-VEGF agent ranibizumab can improve retinal anatomy and vision in eyes with severe CRVO [119]. Despite the significant clinical benefit of anti-VEGF therapy, it does not reduce the risk of neovascular complications; it only delays their onset. Thus, anti-VEGF therapy alone will not control CRVO with severe ischemia. Therefore, while anti-VEGF therapy is working, it may be necessary to perform panretinal photocoagulation; then, once photocoagulation has caused CRVO to revert from ischemic to nonischemic, anti-VEGF therapy should be continued. VEGF appears to be critical in the positive feedback loop, meaning it would be important to inhibit VEGF as soon as possible to block the establishment of positive feedback. Consistent with this, patients with shorter durations of CRVO before initiation of treatment with ranibizumab show greater changes in best-corrected visual acuity from baseline after 12 months than those who had CRVO for longer before receiving treatment [120], suggesting that early anti-VEGF therapy blocks the positive feedback loop. According to our hypothesis, this is because ranibizumab inhibits VEGF, leading to a decrease in leukocyte rolling and adhesion (leukostasis) that prevents the positive feedback loop from being established. Moreover, in the CRUISE trial, which investigated the efficacy and safety of ranibizumab injections in patients with macular edema secondary to CRVO, the visual prognosis was poor when anti-VEGF therapy was delayed [121]. Therefore, anti-VEGF therapy should be started as early as possible to block positive feedback and prevent macular edema from developing into a chronic and refractory condition. We propose that anti-VEGF therapy inhibits the positive feedback loop because it reduces the levels of downstream inflammatory cytokines such as MCP-1 and IL-8, as well as VEGF itself [122], leading to an improvement of ischemia and inflammation. In a study of monthly ranibizumab treatment for CRVO, retinal ischemia increased in the placebo group but not the active drug group, with a significant difference at 6 months. After 6 months, patients in the placebo group also received intravitreal ranibizumab, and the percentage of patients without retinal nonperfusion increased until there was no longer any difference between the two groups [123]. Thus, aggressive VEGF blockade prevents the progression of retinal nonperfusion and promotes reperfusion and restoration of the BRB by controlling the positive feedback loop [115]. In addition, aqueous flare significantly decreased at 1 and 6 months after anti-VEGF therapy [124], suggesting that this therapy also inhibited inflammation. Accordingly, anti-VEGF therapy seems to improve ischemia and inflammation in patients with CRVO by inhibiting VEGF and inflammatory cytokines to suppress leukocyte chemotaxis and adhesion in the retinal vessels.

## 8. Conclusions

The mechanism underlying the pathogenesis of CRVO is complex. We have reviewed it here with a focus on cytokines, discussing how inflammation and ischemia are promoted by VEGF and other inflammatory cytokines such as MCP-1 and ICAM-1. Anti-VEGF agents are most effective in the early stages of CRVO because the ischemia caused by vascular obstruction leads to an increase in the expression of VEGF. Over time, a positive feedback loop is established (Figure 3), which increases the upregulation of inflammatory cytokine expression and causes increasingly severe inflammation. Alternatively, as discussed above, inflammation may cause macular edema and then CRVO. Eventually, CRVO becomes resistant to anti-VEGF therapy. In addition, as shown in Figure 4, the increase in inflammatory cytokines may lead to leukocyte abnormalities and a reduction in retinal blood flow velocity, meaning that the risk of nonischemic CRVO developing into ischemic CRVO increases. It is unclear whether reduced blood flow causes inflammation or vice versa. Once CRVO is ischemic, the overexpression of VEGF may reach excessive levels and result in neovascular glaucoma.

Although the exact mechanisms through which reduced blood flow and inflammation interact in macular edema and CRVO, we hope that this discussion of CRVO pathogenesis, which is based on the most recent available data, will be of use to researchers and clinicians when considering novel therapeutic approaches.

## Figures and Tables

**Figure 1 jcm-09-03457-f001:**
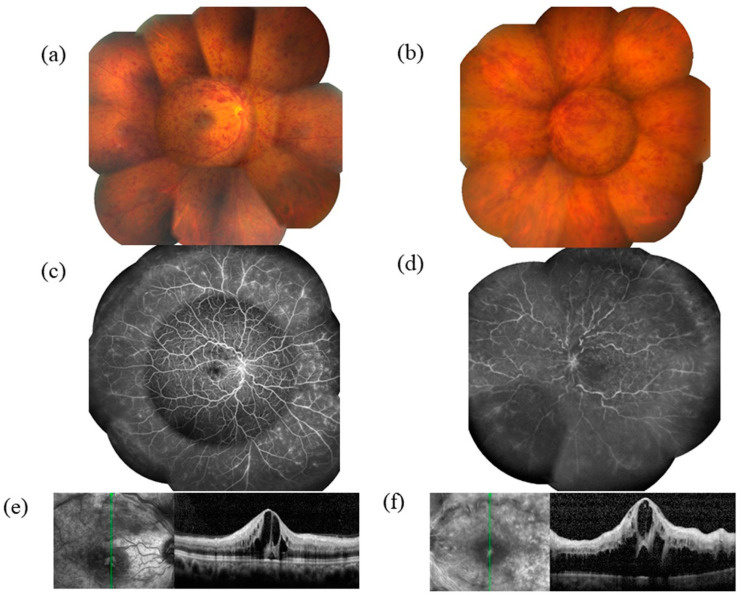
Images of central retinal vein occlusion (CRVO). (**a**) Color fundus photograph of nonischemic CRVO. (**b**) Fluorescein angiogram showing ischemic CRVO. (**c**) Mild retinal hypoxia in a case of nonischemic CRVO. (**d**) Severe retinal hypoxia in a case of ischemic CRVO. (**e**,**f**) Optical coherence tomography images of macular edema and serous retinal detachment in (**e**) a case of nonischemic CRVO (in which they can occur even if retinal hypoxia is mild) and (**f**) a case of ischemic CRVO (in which they are common).

**Figure 2 jcm-09-03457-f002:**
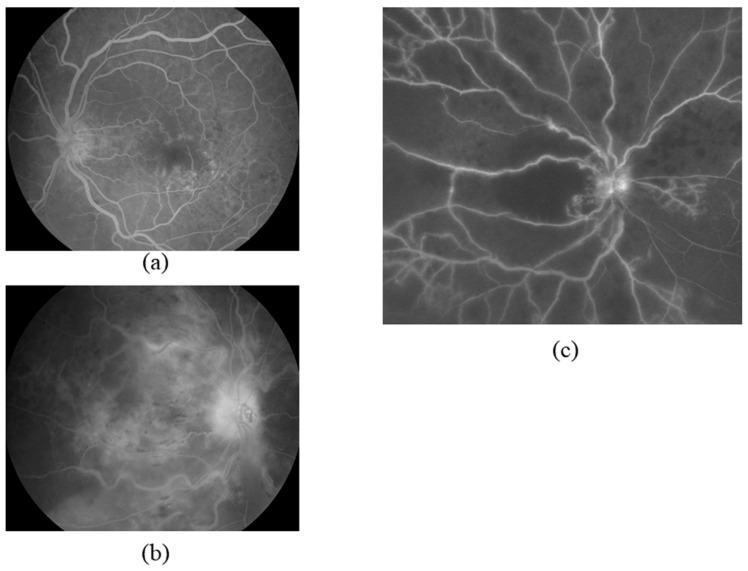
Representative fluorescein angiograms indicating that the severity of ischemia correlates to the concentration of vascular endothelial growth factor (VEGF) in the aqueous humor in cases of retinal nonperfusion. Values shown are aqueous VEGF levels; they are typical of values observed in mild, moderate, and severe cases of ischemia. (**a**) Mild ischemia (41,3 pg/mL). (**b**) Moderate ischemia (327 pg/mL). (**c**) Severe ischemia (1572 pg/mL).

**Figure 3 jcm-09-03457-f003:**
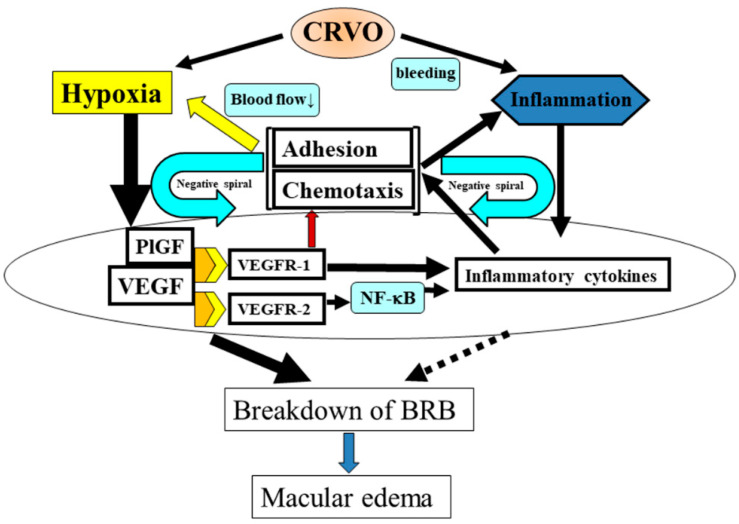
Mechanism of macular edema pathogenesis in central retinal vein occlusion (CRVO). CRVO causes ischemia and produces inflammation that is secondary to impaired retinal perfusion and hemorrhage. This is associated with an upregulation of VEGF and inflammatory cytokine expression, which disrupts the blood–retinal barrier (BRB) and ultimately results in macular edema. VEGF and PlGF activate VEGFR-1, leading to leukocyte recruitment and further increasing inflammatory cytokine expression. In addition, VEGF activates VEGFR-2, which increases vascular permeability and further enhances the expression of inflammatory cytokines such as MCP-1 and ICAM-1 via NF-κB. The resultant chemotaxis and adhesion of leukocytes to the vascular endothelium decreases blood flow velocity. Blood flow velocity also worsens with increasing retinal edema, which results in longer diffusion distances. Decreased blood flow velocity thus creates a positive feedback loop that further exacerbates ischemia. Treatment of edema also compensates for the relative ischemia and leads to less leakage into the tissue. The resulting phagocytosis of leaking material, blood, and ischemic tissue may be an important contributor to maintaining and worsening the inflammation. As a result, the pathological mechanism increases in complexity. This hypothesis predicts that inflammation gains influence in the pathogenesis of CRVO as it develops into a chronic condition.

**Figure 4 jcm-09-03457-f004:**
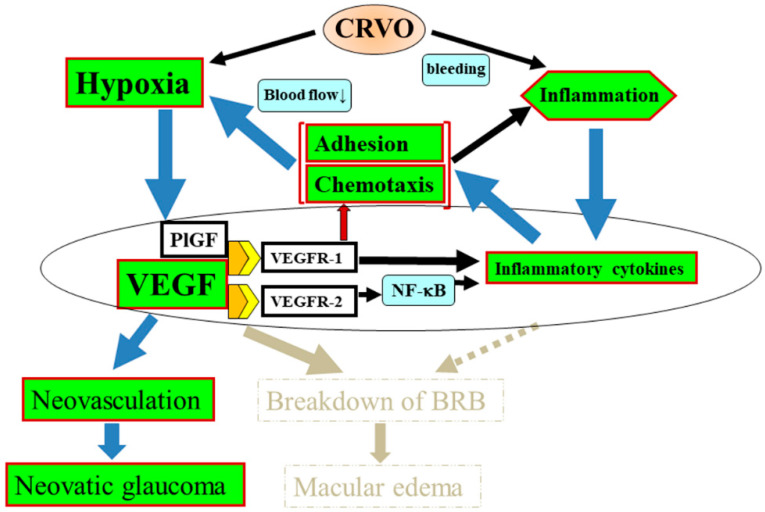
Mechanism of neovascularization pathogenesis in central retinal vein occlusion (CRVO). CRVO is associated with elevated expression of inflammatory cytokines such as MCP-1 and ICAM-1, which leads to leukocyte abnormalities and a decrease in retinal blood flow velocity. The slowing of the blood flow creates a positive feedback loop that further exacerbates ischemia. In addition, the resultant leukocyte chemotaxis and adhesion enhance inflammation, creating a second positive feedback loop. As a result, the risk of CRVO developing from nonischemic to ischemic increases. Finally, hypoxia causes excessive overexpression of VEGF, which results in vitreous hemorrhage and neovascular glaucoma.

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
