# Peer review of "Cytokines and Pathogenesis of Central Retinal Vein Occlusion"

_jcm, 2020, doi:10.3390/jcm9113457_

Round 1

Reviewer 1 Report

jcm-944942

I enjoyed reading the well-written manuscript of H. Noma and colleagues, Though there are some doubts that deserve to be discussed:

Arteriosclerosis and hypertension are age-related and not only linked to lifestyle, which renders this assumption unfair for patients suffering these diseases.

Line 37-38: Given the prevalence of this disease and the options available for its treatment, the dramatic wording of an urgent need is questionable.

Lines 41-44: Please add for the readers the relevant differences between mentioned BRVO and CRVO, since both have been recognised to be capable of inducing severe vision loss and secondary rubeotic glaucoma.

Line 47: “arteriosclerotix”: It is not as simple as that. What about the young pts without any comorbidity?

Line 56-7: Not true, BRVO is less likely to, but may… depending on area and severity of ischemia. Mind the repetition of the aforementioned.

Figure 2, Line 67: “showing” is incorrect. …indicating that the severity of ischemia correlates to VEGF concentrations” would be appropriate.

Line 73: Its expression is wrong, there exists a physiological expression. Re word to “Upregulation of its expression” for correctness.

Lines 83-84: Explain why/how patients could have NO VEGF level increase

Line 92: remove repetition.

Chapter 4: Lines 106-115, 126-146 read well, but the authors might have to add information about the role of these messengers in the retinal microenvironment and how they contribute to its bio-equilibrium.

Lines 158-62: How can that hold true, if anti-VEGFs are  more effective in non-.ischemic RVO as stated in lines 96-9, pls explain.

Line 165: … secondary to impaired retinal perfusion and hemorrhage: Precisely, if true, the degree of retinal hemorrhaging would have to be shown to be closely correlated to the VEGF concentrations, which has still to be demonstrated. I rather think that the degree of hemorrhaging correlates to the degree of acute impairment of microcirculation, anything thereafter correlates to re-perfusion, and this needs to be discussed in the context of RVO as importantly as the degree of ischemia. Early inflammation may be more importantly be kinked to reperfusion than to ischemia.

Line 171: blood flow velocity worsens with increasing retinal edema, which also results in longer diffusion distances which adds to the severity of ischemia. Treatment of edema improves not only edema, but also recompensates relative ischemia, and it leads to less leakage into the tissue,  which in turn needs to be phagocytosed. Phagocytosis of leaking material, blood and ischemic tissue may be important contributors to entertain and worsen the inflammation.

Line 208: …suggesting the presence: yes if the reduced blood flow velocity is the major trigger of inflammation, which is unlikely. This assumption thus is not evident, inflammation could also result from edema, ischemia, phagocytosis etc…

Line 215: ref missing ([87]?)

Line 220: result of retinal ischemia and hemorrhage, not only henmorrhage (see my argumentation above or tightly argue why not).

Lines 230-232: even true in the absence of hemorrhage which adds to the doubts of the role of hemorrhages

Line 263: no, restoration of the BRB

Personally, I fully support the idea of an important role of inflammation, but do not think that retinal hemorrhage is the reason, but consequence of vascular occlusion induced inflammation. Moreover, BRVO and CRVO may principally not be so different I their pathophysiology. The degree of tissue damage, however, is much more readily compensated in BRVO. After all, I congratulate the authors for putting together an interesting and well-written manuscript on the pathogenesis of RVO.

Author Response

Response to Reviewer 2 Comments

I enjoyed reading the well-written manuscript of H. Noma and colleagues, Though there are some doubts that deserve to be discussed:

Point 1: Arteriosclerosis and hypertension are age-related and not only linked to lifestyle, which renders this assumption unfair for patients suffering these diseases.

Response 1: Thank you for your positive remark. We agree with your suggestion and have changed “with lifestyle diseases” to “with diseases” (page 1, lines 11 and 35).

Point 2: Line 37-38: Given the prevalence of this disease and the options available for its treatment, the dramatic wording of an urgent need is questionable.

Response 2: Thank you for useful suggestion. We have changed “This highlights an urgent need …” to “This development highlights the need ...” (page 1, lines 37 and 38).

Point 3: Lines 41-44: Please add for the readers the relevant differences between mentioned BRVO and CRVO, since both have been recognised to be capable of inducing severe vision loss and secondary rubeotic glaucoma.

Response 3:  Thank you for this useful suggestion. We have added text about the relevant differences between the 2 conditions, including the tendency of CRVO to result in more severe symptoms than BRVO (page 2, lines 63-67). Furthermore, we have moved the text on the consequences of BRVO and CRVO to the end of the section so that all information on pathology is in one place (see response to Point 5).

Point 4: Line 47: “arteriosclerotix”: It is not as simple as that. What about the young pts without any comorbidity?

Response 4: Thank you for mentioning this point. We have changed the text and added information, as follows:

“CRVO occurs mainly in older people when the central retinal vein is compressed by an arteriosclerotic central retinal artery in the vicinity of the lamina cribrosa, resulting in venous thrombosis [5]. …………. However, CRVO may also occur in younger people without any comorbidity; the pathogenesis of these cases is unclear, but case reports and series proposed inflammation, exercise and dehydration, and congenital anomalies as potential explanations [6-8].” (page 2, lines 51-54, and 47-49).

New references:

Fong A. C. and Schatz H. (1993) Central retinal vein occlusion in young adults. Surv Ophthalmol 37(6):393-417

Moisseiev E. et al. (2014) Intense Exercise Causing Central Retinal Vein Occlusion in a Young Patient: Case Report and Review of the Literature. Case Rep Ophthalmol 5(1): 116-120

Walters R. F. and Spalton D. J. (1990) Central retinal vein occlusion in people aged 40 years or less: a review of 17 patients. Br J Ophthalmol 74(1): 30-35

Point 5: Line 56-7: Not true, BRVO is less likely to, but may… depending on area and severity of ischemia. Mind the repetition of the aforementioned.

Response 5: Thank you for mentioning this point. We have reworded the text as follows: “Both BRVO and CRVO can cause vitreous hemorrhage and rubeotic glaucoma: These sequelae are more common in CRVO and can lead to blindness [4], but they may also occur in BRVO, depending on the area and severity of ischemia. Nevertheless, generally CRVO has a worse outcome than BRVO [14].” As mentioned in our response to Point 3, to avoid repetition we have moved the text on pathology from the end of the first paragraph to here (page 2, lines 64-67).

New reference:

Kida T. (2017) Mystery of Retinal Vein Occlusion: Vasoactivity of the Vein and Possible Involvement of Endothelin-1. Biomed Res Int 2017:4816527

Point 6: Figure 2, Line 67: “showing” is incorrect. …indicating that the severity of ischemia correlates to VEGF concentrations” would be appropriate.

Response 6: Thank you for the comment. As suggested, we have replaced “showing…” by the text proposed by the reviewer (page 3, lines 77 and 78).

Point 7: Line 73: Its expression is wrong, there exists a physiological expression. Re word to “Upregulation of its expression” for correctness.

Response 7: Thank you for the comment. As suggested, we have replaced “Its expression” by “Upregulation of its expression” (page 3, line 84).

Point 8: Lines 83-84: Explain why/how patients could have NO VEGF level increase

Response 8: Thank you for mentioning this point. We had not reported the findings of Noma et al. [18] accurately, ie, they actually found that VEGF levels were lower in some patients. We have revised the manuscript as follows:

“However, VEGF levels vary widely between patients, and they are significantly lower in patients with CRVO without retinal ischemia than in those with retinal ischemia [18]” (page 3, lines 95-98).

Point 9: Line 92: remove repetition.

Response 9: Thank you for noticing the repetition in this paragraph. We have reworded it, as follows:

“VEGF also increases the vascular permeability of the eye. The level of VEGF increases with the severity of retinal ischemia [13], leading to more neovascularization and potentially resulting in vitreous hemorrhage and neovascular glaucoma” (page 4, lines 105 and 106).

Point 10: Chapter 4: Lines 106-115, 126-146 read well, but the authors might have to add information about the role of these messengers in the retinal microenvironment and how they contribute to its bio-equilibrium.

Response 10: Thank you for this valuable comment. Interestingly, Reviewer 3 made a similar request. We have added some information on the various messengers, as follows:

“In nonischemic CRVO, PlGF may act together with sICAM-1 and IL-8 to reduce the relative blood flow velocity, measured as the mean blur rate (MBR) [34]” (page 4, lines 128 and 129).

“PDGF is involved in the formation of blood vessels and astrocyte recruitment during development of the retina [35, 36].” (page 4, lines 132 and 133).

“In CRVO, the concentration of PDGF-AA is significantly higher in the aqueous humor and PDGF-AA is hypothesized to contribute to inflammation [39]” (page 4, lines 135 and 136).

“MCP-1 correlates with morphologic changes, such as serious retinal detachment in CRVO [45].” (page 4, line 140).

“IL-6 plays a role in protecting retinal ganglion cells [52, 53]; furthermore, it is hypothesized to mediate the inflammation and remodeling of retinal blood vessels induced by angiotensin II and to be involved in axonal degeneration in glaucoma [53, 54]. Patients with CRVO with ischemia had significantly higher vitreous fluid levels of IL-6 than patients with CRVO without ischemia, and IL-6 levels correlated significantly with the extent of macular edema [18].” (page 5, lines 150-154).

“IL-8 is expressed in the retina, is activated in various ocular diseases, including CRVO and glaucoma, and may play an important role in the activation of other cytokines [26, 65, 66]” (page 5, lines 159-161).

“Levels of IP-10 are elevated in the vitreous fluid in retinal vein occlusion, as well as in proliferative diabetic retinopathy and rhegmatogenous retinal detachment [66].” (page 5, lines 167 and 168).

“PTX3 is produced by retinal pigment epithelial cells in response to proinflammatory cytokines and may mediate the retinal inflammatory response [82]. Park et al. [83] found that levels of PTX3 are significantly higher in patients with CRVO and BRVO than in controls and proposed that PTX3 may be a suitable biomarker for diagnosing retinal vein occlusion.” (page 5, lines 179-182).

New references:

Andrae J. et al. (2008) Role of platelet-derived growth factors in physiology and medicine. Genes Dev 22 (10): 1276-1312

Echevarria F. D. et al. (2016) Interleukin-6: A Constitutive Modulator of Glycoprotein 130, Neuroinflammatory and Cell Survival Signaling in Retina. J Clin Cell Immunol 7(4): 439

Echevarria F. D. et al. (2017) Interleukin-6 Deficiency Attenuates Retinal Ganglion Cell Axonopathy and Glaucoma-Related Vision Loss. Front Neurosci 11:318

Edqvist P.-H. D. et al. (2012) Platelet-Derived Growth Factor Over-Expression in Retinal Progenitors Results in Abnormal Retinal Vessel Formation. PLoS One 7(8): e42488

Koss M. J. et al. (2013) Correlation from Undiluted Vitreous Cytokines of Untreated Central Retinal Vein Occlusion with Spectral Domain Optical Coherence Tomography. The Open Ophthalmology Journal 7:11-17

Kuchtey J. et al. (2010) Multiplex Cytokine Analysis Reveals Elevated Concentration of Interleukin-8 in Glaucomatous Aqueous Humor. Invest Ophthalmol Vis Sci 51(12): 6441-6447

Noma H. et al. (2020) Relationship between retinal blood flow and cytokines in central retinal vein occlusion. BMC Ophthalmology 20, 215

Park K. S. et al. (2014) Elevated Plasma Pentraxin 3 and Its Association with Retinal Vein Occlusion. Korean J Ophthalmol 28(6): 460-465

Rojas M. et al. (2010) Role of IL-6 in angiotensin II-induced retinal vascular inflammation. Investigative Ophthalmology & Visual Science 51(3): 1709-1718

Takahashi S. et al. (2016) Profiles of Inflammatory Cytokines in the Vitreous Fluid from Patients with Rhegmatogenous Retinal Detachment and Their Correlations with Clinical Features. Biomed Res Int 2016:4256183

Woo J. M. et al. (2013) Human retinal pigment epithelial cells express the long pentraxin PTX3. Mol Vis 19:303-310

Point 11: Lines 158-62: How can that hold true, if anti-VEGFs are more effective in non-ischemic RVO as stated in lines 96-9, pls explain.

Response 11: Thank you for this valuable comment. Because ischemic CRVO causes severe cytopathy, the prognosis for regaining visual acuity is poor, even if anti-VEGF therapy is administered. Also, VEGF levels are not particularly high in nonischemic CRVO, and VEGF can be suppressed by anti-VEGF therapy. We have added this point to the manuscript (page 4, lines 113-116).

Point 12: Line 165: … secondary to impaired retinal perfusion and hemorrhage: Precisely, if true, the degree of retinal hemorrhaging would have to be shown to be closely correlated to the VEGF concentrations, which has still to be demonstrated. I rather think that the degree of hemorrhaging correlates to the degree of acute impairment of microcirculation, anything thereafter correlates to re-perfusion, and this needs to be discussed in the context of RVO as importantly as the degree of ischemia. Early inflammation may be more importantly be linked to reperfusion than to ischemia.

Response 12: We agree with the reviewer’s comment. We have changed “secondary to retinal hemorrhage” to “secondary to impaired retinal perfusion and hemorrhage” (page 6, line 201).

Point 13: Line 171: blood flow velocity worsens with increasing retinal edema, which also results in longer diffusion distances which adds to the severity of ischemia. Treatment of edema improves not only edema, but also recompensates relative ischemia, and it leads to less leakage into the tissue, which in turn needs to be phagocytosed. Phagocytosis of leaking material, blood and ischemic tissue may be important contributors to entertain and worsen the inflammation.

Response 13: Thank you for providing these details. We have rewritten the sentence as follows:

“Blood flow velocity also worsens with increasing retinal edema, which results in longer diffusion distances. Decreased blood flow velocity thus creates a positive feedback loop that further exacerbates ischemia. Treatment of edema also compensates for the relative ischemia and leads to less leakage into the tissue. The resulting phagocytosis of leaking material, blood, and ischemic tissue may be an important contributor to maintaining and worsening the inflammation.” (page 6, lines 208-213).

Point 14: Line 208: …suggesting the presence: yes if the reduced blood flow velocity is the major trigger of inflammation, which is unlikely. This assumption thus is not evident, inflammation could also result from edema, ischemia, phagocytosis etc…

Response 14: Thank you for this comment. We have deleted the assumption from the sentence, as follows:

“We compared retinal blood flow velocity in healthy individuals and patients with BRVO or CRVO and found that the more severe the retinal vein occlusion, the lower the blood flow velocity [103]” (page 7, lines 247-250).

Point 15: Line 215: ref missing ([87]?)

Response 15: Thank you for pointing out this mistake. We have added the reference (page 7, line 256).

Point 16: Line 220: result of retinal ischemia and hemorrhage, not only henmorrhage (see my argumentation above or tightly argue why not).

Response 16: We agree with the reviewer and have reworded the sentence as follows:

“In summary, we propose that CRVO causes inflammation as a result of retinal ischemia and hemorrhage, which increases the expression of VEGF and other inflammatory factors in the eye and disrupts the BRB, thus resulting in the onset and progression of macular edema” (page 7, lines 264-267).

Point 17: Lines 230-232: even true in the absence of hemorrhage which adds to the doubts of the role of hemorrhages

Response 17: Thank you for this interesting comment. We agree with the reviewer’s argumentation. We have added information on hemorrhage, as follows:

“This mechanism includes macular edema becoming refractory as it develops into chronic macular edema as the hemorrhaged blood is gradually absorbed; …” (page 7, lines 275-277).

Point 18: Line 263: no, restoration of the BRB

Response 18:  Thank you for this valuable comment. As suggested, restoration of the BRB has been added to the manuscript (page 8, line 311).

Point 19: Personally, I fully support the idea of an important role of inflammation, but do not think that retinal hemorrhage is the reason, but consequence of vascular occlusion induced inflammation. Moreover, BRVO and CRVO may principally not be so different I their pathophysiology. The degree of tissue damage, however, is much more readily compensated in BRVO. After all, I congratulate the authors for putting together an interesting and well-written manuscript on the pathogenesis of RVO.

Response 19:  Thank you for this valuable comment. As suggested in points 3, 5, and 12, we have revised the manuscript.

Reviewer 2 Report

The present review of Noma et al aims to discuss and review the role of cytokines in the pathogenesis of central retinal vein occlusion. The authors review the current literature of CRVO pathogenesis with a special focus on VEGF and other cytokines. The manuscript nicely addresses several aspects of CRVO, although there are certain issues that should be reconsidered:

Section Pathogenesis of CRVO:

The review starts with a comparison with BRVO (as authors previously reviewed that topic). This should be avoided as readers may have not read their previous work. Authors should start by describing CRVO itself. Afterwards they can compare it with BRVO, but not before defining and describing CRVO.

Editing and language should be amended.

The pathogenesis of ME is not only the result of BRB disruption. There are other aspects involved in ME such as hydrostatic and osmotic pressure that also leads to fluid accumulation and tissue swelling.

Section VEGF and Macular Edema:

Authors should also mention here the angiogenic properties of VEGF (not only that it increases permeability) and discuss them a bit further.

Section Inflammation

This section should be rewritten. Authors simply mention that there are several cytokines altered in CRVO and then they just define their roles. Instead, authors should specify each of the mentioned cytokines and discuss their involvement in CRVO (discuss in which scenarios they are altered, whether they are affected by treatment, if they play a role at systemic or intraocular level, findings in vitro and/or in vivo…)

Authors should discuss further the role of steroids as a treatment option for CRVO.

Authors mention the role of PTX3. Is there any evidence of CRP being involved in CRVO as well?

Section VEGF receptors

Authors mention that PIGF stimulate tissue factor production. It would be nice to specify what TF does.

Section Retinal Blood Flow

There are some concerns regarding this section. Authors claim that leukocyte adhesion and inflammation produce a reduction in blood flow. However, there is evidence that the cause is the other way: i.e. a reduction in blood flow (stagnation and very low shear stress) induces inflammation via KLFs, NFKb, etc. Even in retinal endothelial cells there are several reports showing that static flow induces inflammation, leukocyte adhesion and barrier disruption. Authors should also discuss these mechanisms.

Similarly, they further suggest in the Conclusions that CRVO causes inflammation and ME. But it could be either a positive feedback loop or even that inflammation caused ME and then CRVO.

Language editing should be amended.

Author Response

Response to Reviewer 3 Comments

The present review of Noma et al aims to discuss and review the role of cytokines in the pathogenesis of central retinal vein occlusion. The authors review the current literature of CRVO pathogenesis with a special focus on VEGF and other cytokines. The manuscript nicely addresses several aspects of CRVO, although there are certain issues that should be reconsidered:

Section Pathogenesis of CRVO:

Point 1: The review starts with a comparison with BRVO (as authors previously reviewed that topic). This should be avoided as readers may have not read their previous work. Authors should start by describing CRVO itself. Afterwards they can compare it with BRVO, but not before defining and describing CRVO.

Response 1: Thank you for useful suggestion. We have changed the text and now first describe CRVO and then compare it with BRVO (page 2, lines 63-67).

Point 2: Editing and language should be amended.

Response 2: The manuscript has been edited by a native English speaker.

Point 3: The pathogenesis of ME is not only the result of BRB disruption. There are other aspects involved in ME such as hydrostatic and osmotic pressure that also leads to fluid accumulation and tissue swelling.

Response 3: As suggested, we have added the following sentence:

“Other aspects are involved in macula edema, such as hydrostatic and osmotic pressure, which also lead to fluid accumulation and tissue swelling [10].” (page 2, lines 57-59).

New reference:

Cunha-Vaz J. (2017) Mechanisms of Retinal Fluid Accumulation and Blood-Retinal Barrier Breakdown. Dev Ophthalmol 58:11-20

Section VEGF and Macular Edema:

Point 4: Authors should also mention here the angiogenic properties of VEGF (not only that it increases permeability) and discuss them a bit further.

Response 4: Thank you for this useful suggestion. We have added text discussing VEGF and angiogenesis (page 3, lines 85-87, 94, and 95).

Section Inflammation

Point 5: This section should be rewritten. Authors simply mention that there are several cytokines altered in CRVO and then they just define their roles. Instead, authors should specify each of the mentioned cytokines and discuss their involvement in CRVO (discuss in which scenarios they are altered, whether they are affected by treatment, if they play a role at systemic or intraocular level, findings in vitro and/or in vivo…)

Response 5: Thank you for this suggestion. Reviewer 2 also requested that we add additional information on these inflammatory markers. Unfortunately, we did not have sufficient time to research each cytokine in the depth requested by the reviewer. Nevertheless, we have added some information that we think will be useful to readers, as follows:

“In nonischemic CRVO, PlGF may act together with sICAM-1 and IL-8 to reduce the relative blood flow velocity, measured as the mean blur rate (MBR) [34]” (page 4, lines 128 and 129).

“PDGF is involved in the formation of blood vessels and astrocyte recruitment during development of the retina [35, 36]” (page 4, lines 132 and 133).

“In CRVO, the concentration of PDGF-AA is significantly higher in the aqueous humor and PDGF-AA is hypothesized to contribute to inflammation [39]” (page 4, lines 135 and 136).

“MCP-1 correlates with morphologic changes, such as serious retinal detachment in CRVO [45]” (page 4, line 140).

“IL-6 plays a role in protecting retinal ganglion cells [52, 53]; furthermore, it is hypothesized to mediate the inflammation and remodeling of retinal blood vessels induced by angiotensin II and to be involved in axonal degeneration in glaucoma [53, 54]. Patients with CRVO with ischemia had significantly higher vitreous fluid levels of IL-6 than patients with CRVO without ischemia, and IL-6 levels correlated significantly with the extent of macular edema [18].” (page 4 and 5, lines 150-154).

“IL-8 is expressed in the retina, is activated in various ocular diseases, including CRVO and glaucoma, and may play an important role in the activation of other cytokines [26, 65, 66]” (page 5, lines 159-161).

“Levels of IP-10 are elevated in the vitreous fluid in retinal vein occlusion, as well as in proliferative diabetic retinopathy and rhegmatogenous retinal detachment [66].” (page 5, lines 167 and 168).

“PTX3 is produced by retinal pigment epithelial cells in response to proinflammatory cytokines and may mediate the retinal inflammatory response [82]. Park et al. [83] found that levels of PTX3 are significantly higher in patients with CRVO and BRVO than in controls and proposed that PTX3 may be a suitable biomarker for diagnosing retinal vein occlusion.” (page 5, lines 179-182).

New references:

Andrae J. et al. (2008) Role of platelet-derived growth factors in physiology and medicine. Genes Dev 22 (10): 1276-1312

Echevarria F. D. et al. (2016) Interleukin-6: A Constitutive Modulator of Glycoprotein 130, Neuroinflammatory and Cell Survival Signaling in Retina. J Clin Cell Immunol 7(4): 439

Echevarria F. D. et al. (2017) Interleukin-6 Deficiency Attenuates Retinal Ganglion Cell Axonopathy and Glaucoma-Related Vision Loss. Front Neurosci 11:318

Edqvist P.-H. D. et al. (2012) Platelet-Derived Growth Factor Over-Expression in Retinal Progenitors Results in Abnormal Retinal Vessel Formation. PLoS One 7(8): e42488

Koss M. J. et al. (2013) Correlation from Undiluted Vitreous Cytokines of Untreated Central Retinal Vein Occlusion with Spectral Domain Optical Coherence Tomography. The Open Ophthalmology Journal 7:11-17

Kuchtey J. et al. (2010) Multiplex Cytokine Analysis Reveals Elevated Concentration of Interleukin-8 in Glaucomatous Aqueous Humor. Invest Ophthalmol Vis Sci 51(12): 6441-6447

Noma H. et al. (2020) Relationship between retinal blood flow and cytokines in central retinal vein occlusion. BMC Ophthalmology 20, 215

Park K. S. et al. (2014) Elevated Plasma Pentraxin 3 and Its Association with Retinal Vein Occlusion. Korean J Ophthalmol 28(6): 460-465

Rojas M. et al. (2010) Role of IL-6 in angiotensin II-induced retinal vascular inflammation. Investigative Ophthalmology & Visual Science 51(3): 1709-1718

Takahashi S. et al. (2016) Profiles of Inflammatory Cytokines in the Vitreous Fluid from Patients with Rhegmatogenous Retinal Detachment and Their Correlations with Clinical Features. Biomed Res Int 2016:4256183

Woo J. M. et al. (2013) Human retinal pigment epithelial cells express the long pentraxin PTX3. Mol Vis 19:303-310

Point 6: Authors should discuss further the role of steroids as a treatment option for CRVO.

Response 6: Thank you for useful suggestion. We have added that corticosteroids may suppress positive feedback and prevent the transition from nonischemic to ischemic CRVO (page 8, lines 282 and 283).

Point 7: Authors mention the role of PTX3. Is there any evidence of CRP being involved in CRVO as well?

Response 7: Thank you for useful suggestion. As mentioned in our response to point 5, we have added the role of PTX3 (page 5, lines 179-182). We have also added that higher CRP levels are more likely to be found in CRVO (page 5, lines 171 and 172).

Section VEGF receptors

Point 8: Authors mention that PIGF stimulate tissue factor production. It would be nice to specify what TF does.

Response 8: Thank you for this suggestion. We have added the following text:

“Tissue factor is involved in activating the extrinsic coagulation cascade pathway [92].” (page 6, line 228).

New reference:

Deobhakta A. and Chang L. K. (2013) Inflammation in Retinal Vein Occlusion. Int J Inflam 2013:438412

Section Retinal Blood Flow

Point 9: There are some concerns regarding this section. Authors claim that leukocyte adhesion and inflammation produce a reduction in blood flow. However, there is evidence that the cause is the other way: i.e. a reduction in blood flow (stagnation and very low shear stress) induces inflammation via KLFs, NFKb, etc. Even in retinal endothelial cells there are several reports showing that static flow induces inflammation, leukocyte adhesion and barrier disruption. Authors should also discuss these mechanisms.

Response 9:  Thank you for this comment, which is similar to one made by Reviewer 2. In response to these comments and suggestions, we have changed the legend of Figure 3, as follows:

“Blood flow velocity also worsens with increasing retinal edema, which results in longer diffusion distances. Decreased blood flow velocity thus creates a positive feedback loop that further exacerbates ischemia. Treatment of edema also compensates for the relative ischemia and leads to less leakage into the tissue. The resulting phagocytosis of leaking material, blood, and ischemic tissue may be an important contributor to maintaining and worsening the inflammation.” (page 6, lines 208-213).

Point 10: Similarly, they further suggest in the Conclusions that CRVO causes inflammation and ME. But it could be either a positive feedback loop or even that inflammation caused ME and then CRVO.

Response 10: Thank you for this comment, which is similar to a comment made by Reviewer 2. In addition to our changes to the legend of Figure 3 mentioned above, we have added the following text to this section:

“Blood flow velocity also worsens as retinal edema increases, resulting in longer diffusion distances and creating a positive feedback loop that further worsens ischemia. An alternative hypothesis is that inflammation itself causes macular edema and that CRVO is a consequence of this edema” (page 7, lines 259-262).

Point 11: Language editing should be amended.

Response 11: The manuscript has been proofread by a native English speaker.

Round 2

Reviewer 1 Report

Thanks for a profound revision of this manuscript which contributed to a significant improvement.

Author Response

Response to Reviewer 2 Comments

Point 1: Thanks for a profound revision of this manuscript which contributed to a significant improvement.

Response 1: Thank you very much for your favorable review of our manuscript.

Reviewer 2 Report

Authors have partially addressed some of the suggestions raised by the reviewer. However, there are two sections that have not been fully addressed.

Section Inflammation: Authors have added some comments regarding the role of certain cytokines in CRVO. However, they should discuss further the current literature regarding such mediators and CRVO (clinical, in vivo, and in vitro studies). To attract the reader’s attention, it would be better to start each cytokine discussing their clinical role in CRVO (eg. Cytokine xxx is a proinflammatory mediator that is increased in serum with patients with CRVO…. And afterwards elaborate more on the role of such cytokine.

Section Retinal Blood Flow: There are still concerns regarding this section. Authors claim that leukocyte adhesion and inflammation produce a reduction in blood flow. However, there is evidence that the cause is the other way around: i.e. a reduction in blood flow (stagnation and very low shear stress) induces inflammation via KLFs, NFKb, etc. Even in retinal endothelial cells there are several reports showing that static flow induces inflammation, leukocyte adhesion and barrier disruption. Authors should also discuss these mechanisms. As in the revised version they only mention the possibility of the positive feedback loop. They should discuss how reduced blood flow activates and induces a proinflammatory state.

Similarly, they further suggest in the Conclusions that CRVO causes inflammation and ME. But it could be either a positive feedback loop or even that inflammation caused ME and then CRVO.

Author Response

Response to Reviewer 3 Comments

Authors have partially addressed some of the suggestions raised by the reviewer. However, there are two sections that have not been fully addressed.

Point 1: Section Inflammation: Authors have added some comments regarding the role of certain cytokines in CRVO. However, they should discuss further the current literature regarding such mediators and CRVO (clinical, in vivo, and in vitro studies). To attract the reader’s attention, it would be better to start each cytokine discussing their clinical role in CRVO (eg. Cytokine xxx is a proinflammatory mediator that is increased in serum with patients with CRVO…. And afterwards elaborate more on the role of such cytokine.

Response 1:

Thank you for your suggestion. We have added text to the section on inflammation (“4. Inflammation and other putative mechanisms of macular edema in CRVO”). To emphasize the importance of cytokines, we now discuss them at the beginning of this section. As you requested, we have added additional cytokines and discuss the current literature regarding their role in CRVO; the discussions include information on clinical, in vivo, and in vitro studies. Furthermore, to make this section easier for readers to navigate we have divided it into sections and added subheadings. We have also added literature and updated the reference list (pages 4-6, lines 115, 116, 119-139, 141-150, 153-155, 157-161, 163-168, 176-190, 192-195, 197-201, 204-212, and 214-233).

Point 2: Section Retinal Blood Flow: There are still concerns regarding this section. Authors claim that leukocyte adhesion and inflammation produce a reduction in blood flow. However, there is evidence that the cause is the other way around: i.e. a reduction in blood flow (stagnation and very low shear stress) induces inflammation via KLFs, NFKb, etc. Even in retinal endothelial cells there are several reports showing that static flow induces inflammation, leukocyte adhesion and barrier disruption. Authors should also discuss these mechanisms. As in the revised version they only mention the possibility of the positive feedback loop. They should discuss how reduced blood flow activates and induces a proinflammatory state.

Response 2:

We agree with your opinion that reduced blood flow may induce inflammation. Therefore, we have added text on this topic to the section on inflammation and reduced blood flow velocity (page 8, lines 309-314, 317, and 318).

Point 3: Similarly, they further suggest in the Conclusions that CRVO causes inflammation and ME. But it could be either a positive feedback loop or even that inflammation caused ME and then CRVO.

Response 3:

We agree that the opposite mechanism may also be the underlying cause of CRVO. Therefore, we have revised our discussion in the conclusion about the various hypothetical interactions between inflammatory events, reduced blood flow, macular edema, and CRVO (page 10, lines 383, 386-388, 390, 391, 393, and 394).
